METHODS AND RESOURCES

# MEANtools integrates multi-omics data to identify metabolites and predict biosynthetic pathways

Kumar Saurabh Singh[1,2,3,4*], Hernando Suarez Duran[1], Elena Del Pup[1], Olga Zafra Delgado[5], Saskia C. M. van Wees[2], Justin J. J. van der Hooft[1,6*], Marnix H. Medema [1*]

1 Bioinformatics Group, Wageningen University, Wageningen, The Netherlands, 2 Plant–Microbe Interactions, Institute of Environmental Biology, Utrecht University, Utrecht, The Netherlands, 3 Plant Functional Genomics, Brightlands Future Farming Institute, Maastricht University, Maastricht, The Netherlands, 4 Faculty of Environment, Science and Economy, University of Exeter, Exeter, United Kingdom, 5 Departamento de Genética Molecular de Plantas, Centro Nacional de Biotecnología–Consejo Superior de Investigaciones Científicas, Campus Universidad Autónoma, Madrid, Spain, 6 Department of Biochemistry, University of Johannesburg, Auckland Park, Johannesburg, South Africa

* kumarsaurabh.singh@maastrichtuniversity.nl (KSS); justin.vanderhooft@wur.nl (JJJvdH); marnix.medema@wur.nl (MHM)

## Abstract

During evolution, plants have developed the ability to produce a vast array of specialized metabolites, which play crucial roles in helping plants adapt to different environmental niches. However, their biosynthetic pathways remain largely elusive. In the past decades, increasing numbers of plant biosynthetic pathways have been elucidated based on approaches utilizing genomics, transcriptomics, and metabolomics. These efforts, however, are limited by the fact that they typically adopt a target-based approach, requiring prior knowledge. Here, we present MEANtools, a systematic and unsupervised computational integrative omics workflow to predict candidate metabolic pathways de novo by leveraging knowledge of general reaction rules and metabolic structures stored in public databases. In our approach, possible connections between metabolites and transcripts that show correlated abundance across samples are identified using reaction rules linked to the transcript-encoded enzyme families. MEANtools thus assesses whether these reactions can connect transcript-correlated mass features within a candidate metabolic pathway. We validate MEANtools using a paired transcriptomic-metabolomic dataset recently generated to reconstruct the falcarindiol biosynthetic pathway in tomato. MEANtools correctly anticipated five out of seven steps of the characterized pathway and also identified other candidate pathways involved in specialized metabolism, which demonstrates its potential for hypothesis generation. Altogether, MEANtools represents a significant advancement to integrate multi-omics data for the elucidation of biochemical pathways in plants and beyond.

**Data availability statement:** The authors confirm that all data underlying the findings are fully available without restriction. Raw paired-transcriptomics and -metabolomics data for the case study was taken from NCBI BioProject: PRJNA509154 and EBI's MetaboLights: MTBLS1039, respectively. Pre-processed file of the metabolomics data is available at https://github.com/satte-ly-lab/falcarindiol_pathway_metabolomics. Pre-processed RNA-seq expression matrix together with additional files required by MEANtools are available at https://zenodo.org/records/15697913. The additional files include an SQLite formatted LOTUS database, RetroRules-based PFAMs cross-referenced with Rhea and the KEGG-orthology database, a PFAM-EC mapping file, and outputs of the MEANtools correlation analysis for the bacterial and fungal treatment groups in the SQLite database format. MEANtools is open source and is freely available on its GitHub page https://github.com/kumarsaurabh20/meantools, under the permissive MIT license. An initial public release of MEANtools v1.0.0 is available at https://doi.org/10.5281/zenodo.15720912. The MEANtools documentation and tutorial with the demo data is available on GitHub at https://meantools.readthedocs.io/en/latest/.

**Funding:** This work was supported by the Netherlands Organization for Scientific Research (NWO) under the Groot grant [OCENW.GROOT.2019.063 to SvW and MHM, supporting KSS], Vidi grant VI.Vidi.213.183 to MHM (supporting EdP) and Veni grant 863.15.002 to MHM. The funders had no role in study design, data collection and analysis, decision to publish, or preparation of the manuscript.

**Competing interests:** I have read the journal's policy and the authors of this manuscript have the following competing interests: JJJvdH is currently member of the Scientific Advisory Board of NAICONS Srl., Milano, Italy, and consults for Corteva Agriscience, Indianapolis, IN, USA. M.H.M. is a member of the scientific advisory boards of Hexagon Bio and Hothouse Therapeutics Ltd. The other authors declare to have no competing interests.

## Introduction

Plants have long been recognized for their ability to produce a variety of chemical compounds, known as specialized metabolites (SM). It is estimated that a total of over 200,000 plant SMs have been reported so far that can be classified into distinct metabolite classes, mainly terpenoids, alkaloids, phenolics, sulphur-containing compounds, and fatty-acid derivatives [1]. Additionally, metabolomics has revealed an extensive plant 'dark matter', in the sense that a major proportion of metabolites are yet structurally unknown. Also, the functions of most plant SMs are largely unexplored, but they are generally regarded as crucial for fitness and survival [2–6]. Humans have harnessed these chemical compounds in various areas, including traditional medicines, pharmaceuticals, cosmetics, and agricultural products. The biosynthesis of SMs, however, often hinges on external triggers and follows specific metabolic pathways, which are largely unknown [7]. This poses a substantial challenge in obtaining, cultivating, and extracting these compounds in quantities suitable for research or commercial production. This lack of knowledge has driven interest in developing new methodologies to predict and identify new metabolic products as well as the enzymes that catalyze their biosynthesis.

In the past decades, along with cost reductions, substantial progress in the generation of high-throughput omics datasets has resulted in increasing numbers of high-quality genome assemblies, transcriptome, metabolome, and enzyme reaction datasets [8]. Moreover, advances in synthetic biology allow the validation of *in silico* analyses in vivo, increasing the rate at which novel SMs and the associated enzymes can be characterized [9]. This has amplified the discovery and characterization of biosynthetic pathways in plants. Reconstructing biosynthetic pathways computationally requires details about genes that encode enzymes catalyzing reactions, as well as the metabolites involved in these processes. Tools such as plantiSMASH [10], PhytoClust [11], and PlantClusterFinder [12] are instrumental in identifying gene clusters that are likely to encode enzymes associated with SM pathways. Yet, many SM pathways in plants do not have their genes chromosomally clustered. Additionally, co-expression analyses can be employed to predict functional associations between genes based on their expression patterns [13,14]. In this direction, advances in high-resolution mass spectrometry, such as FTICR-MS and Orbitrap, have led to the development of tools like MetaNetter [15] that introduced the use of mass shifts through predefined biochemical transformations. Such transformations networks are highly valuable for interpreting mass differences between metabolites as potential reactions. In general, individual omics-based investigations, such as genomics, transcriptomics, or metabolomics, have played pivotal roles in delineating specific metabolic pathways and their correlated metabolic products [16–25]. Nevertheless, despite these advancements, the intricate genetic makeup and functional diversity of plant biosynthetic pathways continue to present a formidable challenge. Specifically, a key limitation to current transcriptome- and metabolome-based pathway discovery strategies is that they require prior knowledge on a compound or enzyme that can be used as 'bait' [26] to identify other compounds and/or enzymes involved in the same pathway. Yet, such prior knowledge may not always be available.

**Abbreviations:** DAG, directed acyclic graph; DR, decay rates; EC, enzyme commission; FCs, functional clusters; HCAA, hydroxy cinnamic acid amide; MAD, Median Absolute Deviation; MR, mutual rank; PAL, phenylalanine ammonia-lyase; PCC, Pearson correlation coefficient; THT, tyramine N-hydroxycinnamoyl transferase.

A promising solution to this limitation may be found in the integrative analysis of genomic, transcriptomic, and metabolomics data. Due to the intricate, cooperative interplay of genes and metabolites in SM biosynthesis, implementing multi-omic approaches ensures a comprehensive perspective on the entire process. Indeed, the inclusion of multiple omics layers has facilitated the discovery of several biosynthetic pathways [27–33]. Multi-omics integration strategies can be broadly separated into four categories: conceptual, statistical, model, and pathway-based. Each strategy presents distinct challenges, and all have been reviewed in detail before, with multiple examples of successful usage [34,35]. Such integrative omics technologies [8] provide new opportunities for systematic, unsupervised multi-omics approaches for untargeted or de novo discovery of pathways involved in the biosynthesis of SMs.

Here, we introduce MEANtools, a computational pipeline that combines statistical- and reaction-rules-based integration strategies. MEANtools implements a mutual rank (MR)-based [14] correlation approach to capture mass features that are highly correlated with biosynthetic genes. Our pipeline makes use of general reaction rules and metabolite structures, stored in public databases like RetroRules [36] and LOTUS [37], to predict putative reactions that either constitute intermediate steps or complete biosynthetic pathways. The workflow enables users to explore the biosynthetic potential associated with identified mass features and formulate specific hypotheses about potential pathways associated with the corresponding metabolites.

## Results

### MEANtools integrates omics data to link transcripts to metabolites

MEANtools integrates mass features from metabolomics data and transcripts from transcriptomics data to predict possible metabolic reactions and thus generates hypotheses that can be prioritized for experimental validation (Fig 1A). Reaching the prediction stage involves several independent steps, including formatting and annotating the input data, thereby ensuring the data is ready for subsequent meaningful analysis. MEANtools then leverages RetroRules [36], a retrosynthesis-oriented database of enzymatic reactions annotated with known and predicted protein domains and enzymes linked to each reaction, to assess whether observed chemical differences between metabolites (inferred from observed mass shifts) can logically be explained by reactions that are known to be catalyzed by transcript-associated protein families (Fig 1B). To identify putative structure annotations for metabolite features, MEANtools matches their masses to LOTUS [37], a comprehensive well-annotated resource of Natural Products, taking into account possible adducts (Fig 1C). MEANtools correlates the expression of genes with co-abundant metabolites across samples in paired transcriptomics and metabolomics experiments, ideally spanning a range of different conditions, tissues, and timepoints. Although the correlation approach has aided the characterization of diverse metabolic processes in plants by reducing the dimensionality of the problem and thus generating a small set of testable hypotheses, it is known to result in a high number of false positive metabolite-transcript associations when used in isolation. As illustrated in Fig 1D, we use a mutual rank-based correlation method that maximizes highly correlated metabolite-transcript associations.

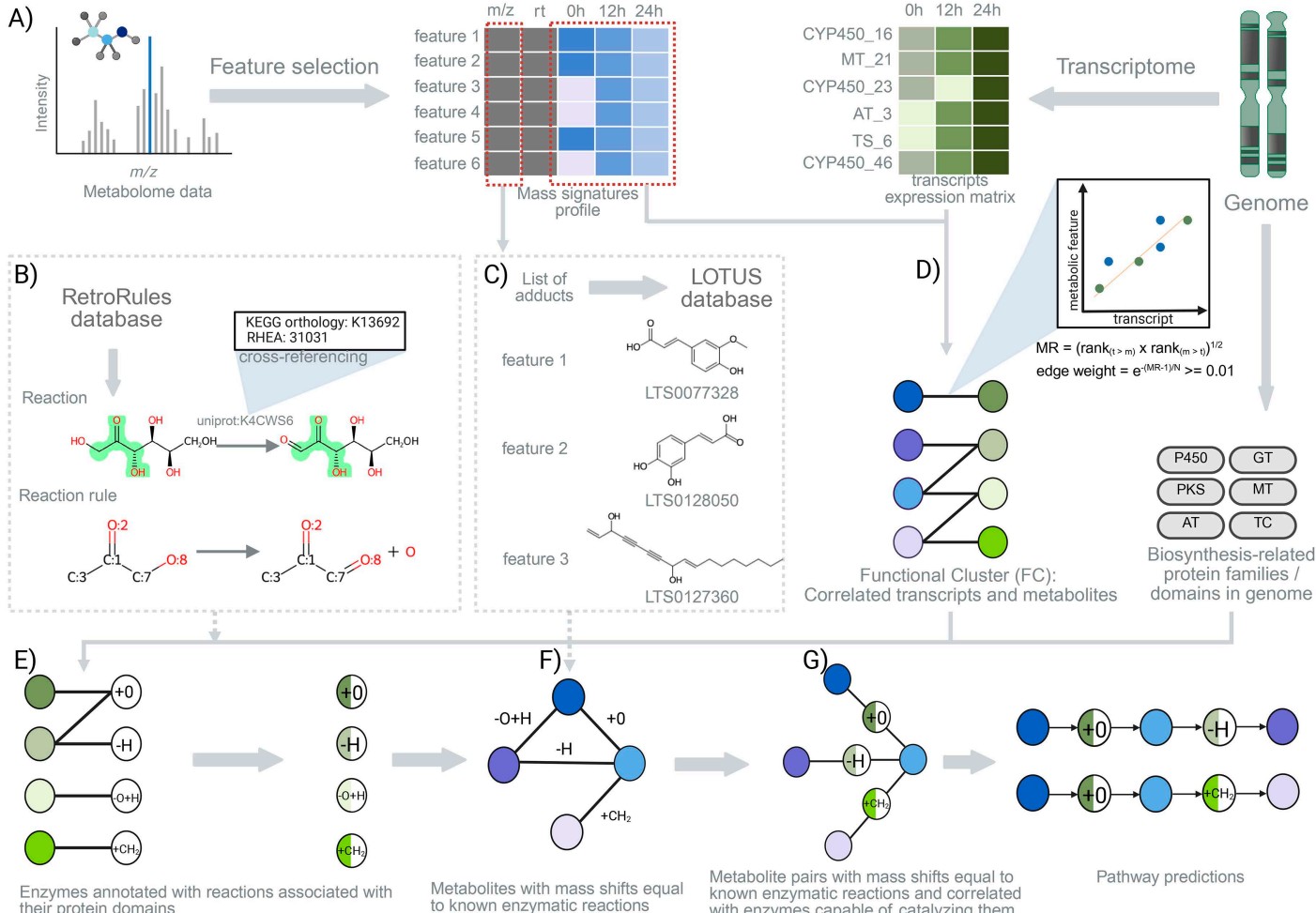

**Fig 1. MEANtools predicts metabolic pathways by integrating transcriptomic, metabolomic, and genomic data. (A)** Mass signature or mass feature profiles are collected using standard metabolomic data processing pipelines. The feature table has rows as unique features and columns are divided into multiple components, like m/z values, retention times, and mass abundance values across samples. Similarly, the transcript expression matrix is collected using a standard RNA-seq data processing pipeline. In the expression matrix, rows represent different transcripts and columns have normalized count data across samples. **(B)** The RetroRules database is formatted by cross-referencing it with the MetaNetX database for its substrate and related mono-isotopic masses. Based on these masses, mass transition values are calculated for all reactions. **(C)** Feature IDs and *m/z* values of mass signatures are mapped against a list of user-defined adducts table. By default, MEANtools provides a list of 48 adducts from both positive and negative mode operations. The list of adducts can be found at https://github.com/kumarsaurabh20/meantools. All m/z values are accounted with the adducts masses and PPM value and mapped against the LOTUS database. This mapping results in the putative annotation of each feature ID with specific structures from the LOTUS database. **(D)** Correlations are computed between expression levels of transcripts and abundances of metabolites. **(E)** The protein families/domains encoded by the genes in the correlated transcript-metabolite pairs are used to query RetroRules and identify which enzymatic reactions may be associated with each transcript. A list of plant-based biosynthetic genes together with their PFAM domain classification can be found in S1 Data. **(F, G)** MEANtools then integrate the results of previous steps to identify cases in which metabolite pairs are correlated to a transcript that encodes an enzyme capable of catalyzing a reaction that explains their mutual mass difference. Finally, MEANtools maps the product of these reactions to other mass signatures in the metabolome and repeats the procedure to generate pathway predictions.

MEANtools then integrates all this information to identify sets of transcript-metabolite pairs that are both highly correlated in abundance and then highlight cases where the metabolites are logically connected by catalytic activities associated with these same transcripts. Thus, MEANtools generates a reaction network where each node is a mass signature within the metabolome, or an unmeasured *ghost mass signature* [38]. In this network, nodes are linked by directed edges

representing enzymatic reactions that can be catalyzed by at least one of the enzyme families encoded by the genes correlated to one of the two mass signatures the reaction links. This network representation of the data allows users to explore the biosynthetic potential of any molecular structure and generate concrete hypotheses about possible pathways leading up to (or from) a given metabolite, which can be tested in the laboratory. Results are displayed in a variety of formats for users to interact with, describing predicted metabolic pathways along with the metabolites, enzymes, and reactions that are potentially involved in them. Altogether, MEANtools serves as a strong basis for the development of methodologies to explore ways in which paired genomic, transcriptomic, and metabolomic data can be used to analyze biosynthetic diversity.

## RetroRules and LOTUS database integration

In the above process, strongly correlated mass feature-transcript pairs are examined using the general reaction rules obtained from the RetroRules. All enzymatic reactions in the RetroRules database are cross-referenced with the Meta-NetX [39], a repository of metabolic networks that MEANtools uses to identify the mass differences (shifts in the masses) between the substrates and products of known enzymatic reactions (Fig 1E). MEANtools then annotates all reactions with an associated mass shift. This step needs to be executed only once, either during the initial retrieval of the database or when it is updated. As a next step, users can manually annotate a subset of mass signatures (mass-to-charge ratios of the measured ions) in the metabolomic dataset with metabolite structures (Fig 1F and 1G). Alternatively, MEANtools can assign potential structure matches by identifying adducts in the metabolome and querying the LOTUS database for matching metabolites based on molecular weight (Fig 1C).

To determine the significance of the presence of experimentally characterized biosynthetic reactions in the RetroRules database, we tested the presence of selected biosynthetic reactions from the Singh and colleagues, review Fig 1 [8] (S2 Data). Among 187 experimentally characterized biochemical reactions, 134 were found in the RetroRules database, and 53 were missing. The presence of 72% of selected reactions in the RetroRules database is significantly higher ($x^2$-statistic: 35.10; DF = 1; $p < 0.001$) than expected under the null hypothesis of equal probability. This indicates that RetroRules database has a good coverage of experimentally characterized biosynthetic reactions, enhancing its reliability for further pathway analysis. Additionally, for the same set of experimentally characterized reactions, we investigated the presence of structures for both the substrates and the products, from the list of experimentally characterized biosynthetic reactions, in the LOTUS database (S2 Data). Compared to the total 374 structures from the selected reactions, 132 structures were found in the database with a significance of $p < 0.001$ ($x^2$-statistic: 32.353; DF = 1), highlighting substantial structural overlap.

RetroRules is populated with ~43,000 reactions annotated with enzymes that are predicted to be associated with all reactions. Most of these annotated enzyme-reaction associations, however, are the result of propagating the annotation of characterized reactions to other reactions with the same enzyme commission (EC) number, and they therefore of various reliability and require verification. To increase confidence in the enzymatic annotations, we cross-referenced each reaction in RetroRules to the manually curated reaction databases Rhea [40] and KEGG [41]. We refined reaction-enzyme associations supported by experimental evidence and then propagated these annotations through KEGG-orthology groups (Methods). This way, we generated three datasets, namely, *strict*, *medium*, and *loose*, differing in the coverage of chemical space and confidence in the enzymatic annotations. This was done to remove the most generic Pfam annotations. *Loose* dataset contains 2,704,948 reaction rules-enzyme associations expanded from the RetroRules database by cross-referencing with the Rhea and KEGG-orthology database (S1 Fig). *Medium* dataset contains 429,267 entries consisting of experimentally validated entries together with the ECDomainMiner predictions. Finally, the *strict* dataset contains 67,501 experimentally validated entries (S1 Fig). These datasets are specifically developed for enzyme function prediction and are especially relevant when specificity is preferred over sensitivity. All three datasets come with taxonomic origin annotations. Users can therefore not only select the datasets between *loose*, *medium*, *strict*, but also use the taxonomy of the samples (S2 Fig) for further refinement of their analyses based on the species-specificity of Pfams.

## Reconstruction of the falcarindiol pathway in tomato

To assess the performance of MEANtools in predicting metabolic pathways, we used data derived from a recently published paired omics dataset. Specifically, we assessed whether MEANtools would be able to reconstruct the falcarindiol pathway in tomato using the dataset published by Jeon *and colleagues* in 2020 [30] in the study that originally elucidated this pathway. MEANtools correctly anticipated five out of seven transformations of intermediate metabolites in the falcarindiol pathway, along with the enzymes that catalyze the reactions. The initial untargeted metabolomics and transcriptomics data comprised 11,266 mass features and 20,576 transcripts. To narrow down the counts and select the most informative mass features and transcripts, we performed differential abundance analysis of mass features and differential expression analysis of transcripts across samples and time-points. After selecting features and transcripts based on a corrected *p*-value and log fold change threshold of 0.01 and 2, respectively, 1,230 mass features and 7,590 transcripts remained. Correlation analysis (step 1), with a minimum absolute *Pearson* correlation coefficient of 0.1, further refined the count of informative mass features and transcripts. Four networks (N) were created with different decay rates (DR). The number of transcripts and mass features assigned to functional clusters (FCs) in N1 (DR = 5) were 2,912 (38.4% of input genes) and 232 (18.9% of input mass features), respectively. Similarly, for N2 (DR = 10) the count was 5,488 (72.3%) and 236 (19.2%). For N3 (DR = 25) and N4 (DR = 50) the count was 6,491 (85.5%)/238 (19.3%), and 6,420 (84.6%)/238 (19.3%), respectively. MEANtools also returns a p-value for every transcript-mass feature correlation. This p-value is based on the hypothesis test whether the true correlation between the two datasets is zero. The distribution of the *p*-values resulting from the correlation step (S3 Fig) is heavily skewed towards the right and significantly (Kolomogorov–Smirnov statistics = 0.987; *p*-value = ~0.0) deviates from what would be expected under the null hypothesis of no significant effects, showing a subset of transcripts and mass features that are significantly associated and reflecting real biological interactions.

In the FCs, we first looked for biosynthetic genes (based on classification using plantiSMASH profile hidden Markov models) predicted to be involved in SM pathways, specifically for falcarindiol-related genes [30]. Our analysis revealed a single FC in N2 encompassing three out of the four biosynthetic genes from this cluster (Fig 2A). This FC, containing all three key biosynthetic genes related to the falcarindiol pathway, also included a CYP450 gene suspected to be involved in the modification of dehydrocrepenynic acid, one of the pathway intermediates within the pathway [30]. Other FCs that harbored mass-features present in Fig 2A were merged and taken further to the pathway prediction step of MEANtools. By using only experimentally validated enzyme-reaction associations (*strict* settings), MEANtools anticipated the second step of the falcarindiol biosynthesis pathway as proposed by Jeon and colleagues (crepenynic acid → dehydrocrepenynic acid), seen in Fig 3A. For this step, MEANtools predicted Solyc12g100250.1, which shows strong correlation (0.744; *p*-value 1.106E-10 (Fig 2D) that Jeon and colleagues identified as a major desaturase in the falcarindiol pathway that was linked to this reaction using transient expression [30]. MEANtools also anticipated steps five and six of the pathways proposed by Jeon and colleagues [30], (i.e., octadecene diynoic acid → octadecadiene diynoic acid → metabolite_6 → metabolite_7), as seen in Fig 3B, and provided candidate genes encoding enzymes with a protein domain that has been characterized as able to perform each reaction. To further explore the predictive power of MEANtools, we repeated the analysis with *medium* and *loose* settings. As we moved from strict to medium and then to *loose* settings, we observed an increase in enzyme associations due to the inclusion of less specific Pfam annotations (S6 Fig). Distribution of the correlation coefficients of all mass feature-transcript associations for the falcarindiol pathway can be seen in S4 Fig. A table with all the predictions is available in S4 Data.

A distinctive strength of MEANtools lies in its ability to predict multi-step metabolic reactions. After assigning each mass feature as substrates and products, MEANtools predicts multiple products for a substrate based on reaction rules. It then transforms this linear data into a directed acyclic graph (DAG), enabling stepwise reaction inference. Cyclic reactions are resolved by removing the weakest transcript-metabolite edge, ensuring that reaction flow proceeds outward (from a substrate root metabolite to multiple predicted product metabolites) from the initial metabolite. The longest valid reaction

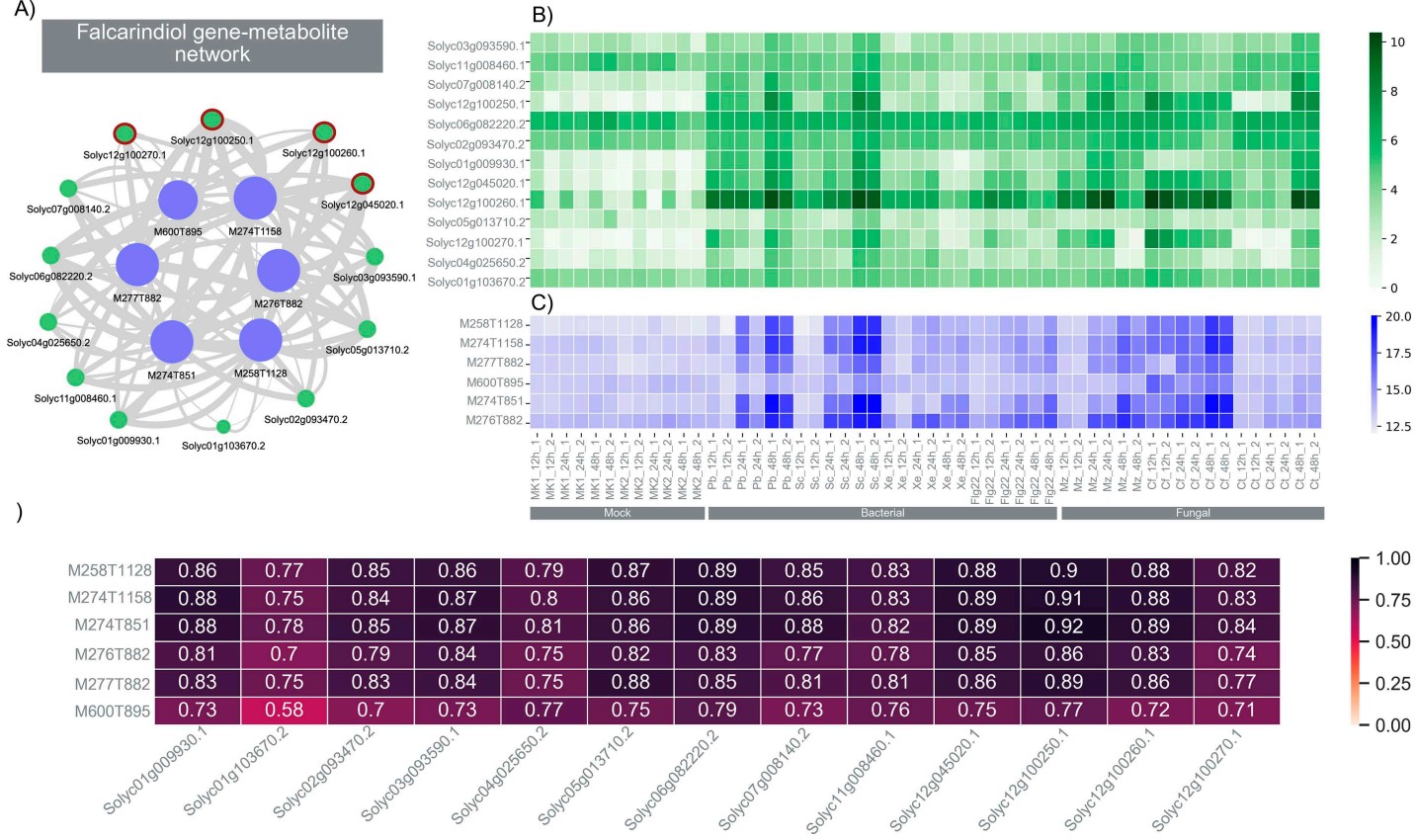

**Fig 2. Identification of the functional cluster (FC) belonging to the falcarindiol pathway. (A)** Network diagram illustrating the connections between transcripts and metabolites within the falcarindiol FC, with pathway-related transcripts marked with an asterisk. The data underlying the network graph can be found at https://zenodo.org/records/15697913. Edge scores values were extracted from the table *Jeon_bac_correlations_MR_weights_DR_10* (Jeon_bacterial.sqlite) for the features present in the table *Jeon_bac_correlations_clone_DR_10 with the* cluster ID number 117. **(B)** Heatmap displaying the expression levels of all genes within the falcarindiol FC. The raw data underlying transcript heatmap can be found at https://zenodo.org/records/15697913/files/rnaseq.rpkm.csv. **(C)** Heatmap showing the abundance of mass-signatures associated with the falcarindiol FC. The raw data underlying heatmap of the falcarindiol-related mass features can be found at https://github.com/sattely-lab/falcarindiol_pathway_metabolomics. **(D)** Summary table presenting the correlations between transcripts and metabolites from the falcarindiol FC. The data underlying the correlation matrix can be found at https://zenodo.org/records/15697913/files/Jeon_bacterial.sqlite in the table *Jeon_bac_correlations*.

path in each DAG is then extracted, allowing MEANtools to anticipate complex biosynthetic routes. This approach empowers MEANtools to predict extended pathway branches, as exemplified in our case study (Fig 3B) where multiple reaction steps are predicted as a single biosynthetic route, starting from *octadecene diyonic acid* up to *falcarindiol*.

## Identification of functional clusters encompassing other tomato metabolic pathways

Within the Jeon *and colleagues*, dataset [30], a wider investigation unveiled multiple FCs housing biosynthetic genes primarily from three distinct metabolic pathways: the hydroxy cinnamic acid amide (HCAA) pathway [42] the α-tomatine pathway [20], and the chlorogenic acid pathway [43].

We identified two FCs containing biosynthetic genes associated with the synthesis of *p*-coumaroyl-CoA from phenylalanine, a process catalyzed by phenylalanine ammonia-lyase (PAL) and 4-coumaric acid co-enzyme A ligase (4CL), as well as the subsequent biosynthesis of *p*-coumaroyltyramine (S9 Fig), a reaction mediated by hydroxycinnamoyl-CoA:tyramine N-hydroxycinnamoyl transferase (THT) (Fig 4). Interestingly, all metabolites within these two FCs were putatively

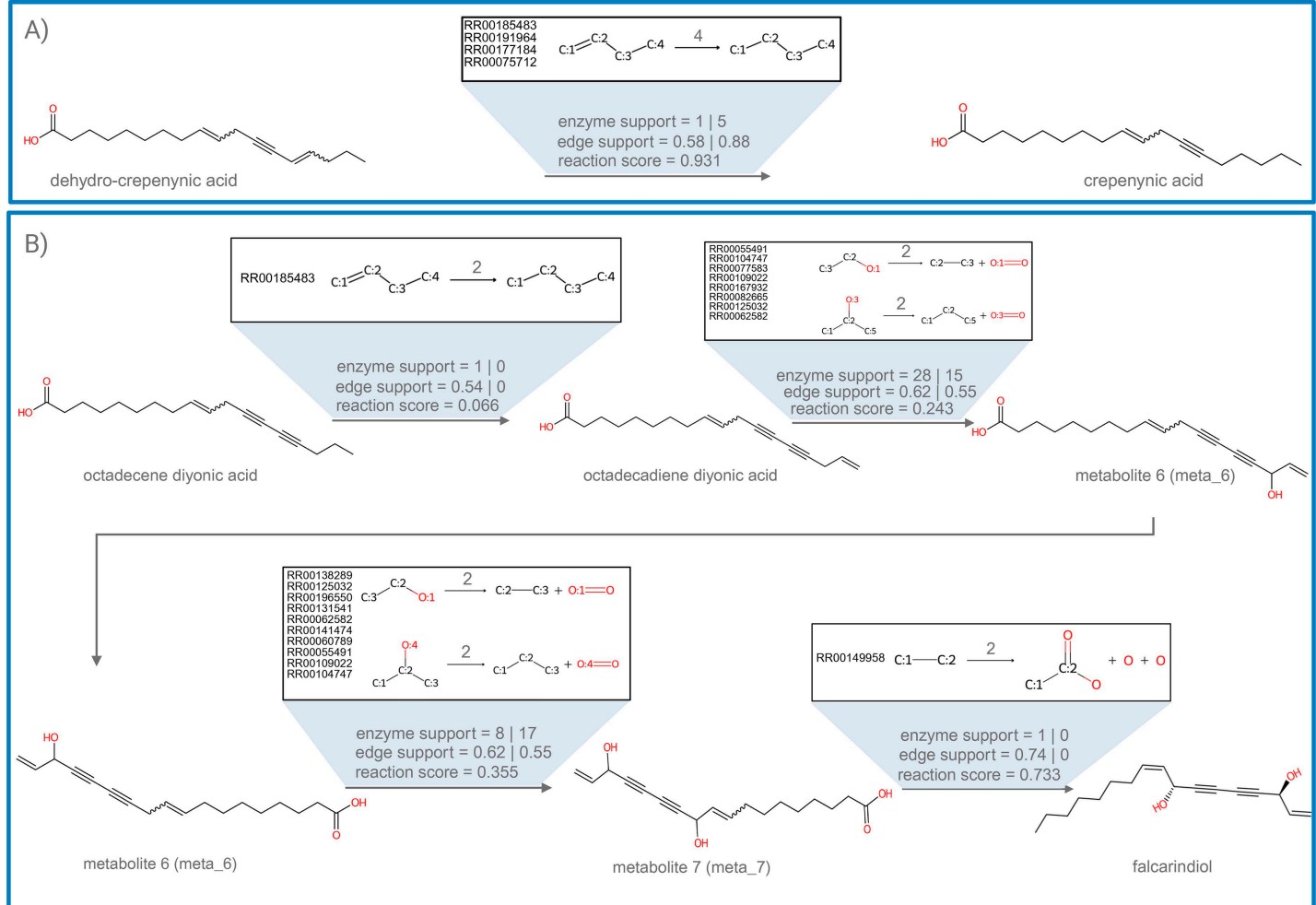

**Fig 3. MEANtools reconstructs parts of the falcarindiol pathway as proposed by Jeon and colleagues [30], and the genes responsible for each enzymatic step. (A)** MEANtools predicts the second step of falcarindiol biosynthesis in reverse (dehydrocrepenynic acid → crepenynic acid). The transformation is annotated with the reaction rule used in the transformation, diameter of reaction and RetroRules-based reaction IDs, enzyme support, edge support, and reaction likelihood. **(B)** MEANtools predicts the third step of falcarindiol biosynthesis in reverse starting from falcarindiol. Each transformation is annotated with a reaction rule associated with that transformation. Additionally, the reaction rule is annotated with the diameter of the reaction and reaction IDs from RetroRules database. Each transformation in the second step of falcarindiol biosynthesis is also annotated with enzyme support, edge support based on correlation values and the reaction likelihood scores. The raw data underlying the predicted reactions and the associated scores can be found at S2 Data.

annotated within the superclass of phenylpropanoids and polyketides (S3 Data). Fig 4A depicts the FC containing PAL (Solyc10g086180; node with a pink border) and 4CL (Solyc03g117870; node with orange border), and metabolites involved in the conversion of phenylalanine to p-coumaroyl-CoA. This FC also contains other co-expressed genes along with PAL and 4CL. Additionally, the correlation analysis performed by MEANtools revealed another FC (Fig 4B) related to the production of p-coumaroyltyramine catalyzed by THT (Solyc08g068790; node with a red border). According to the expression heatmaps depicted in Fig 4C and 4D, while PAL and 4CL showed constitutive expression patterns across both mock and treated samples, THT exhibited significant differential expression ($p$-value < 0.05 and logFC = 4.7) in samples treated with fungal pathogens compared to mock-treated ones. Both the metabolite and genes present in the THT FC (Fig 4B) show overlapping abundance and expression patterns (highlighted with black solid bar in the heatmaps of Fig 4C and 4D).

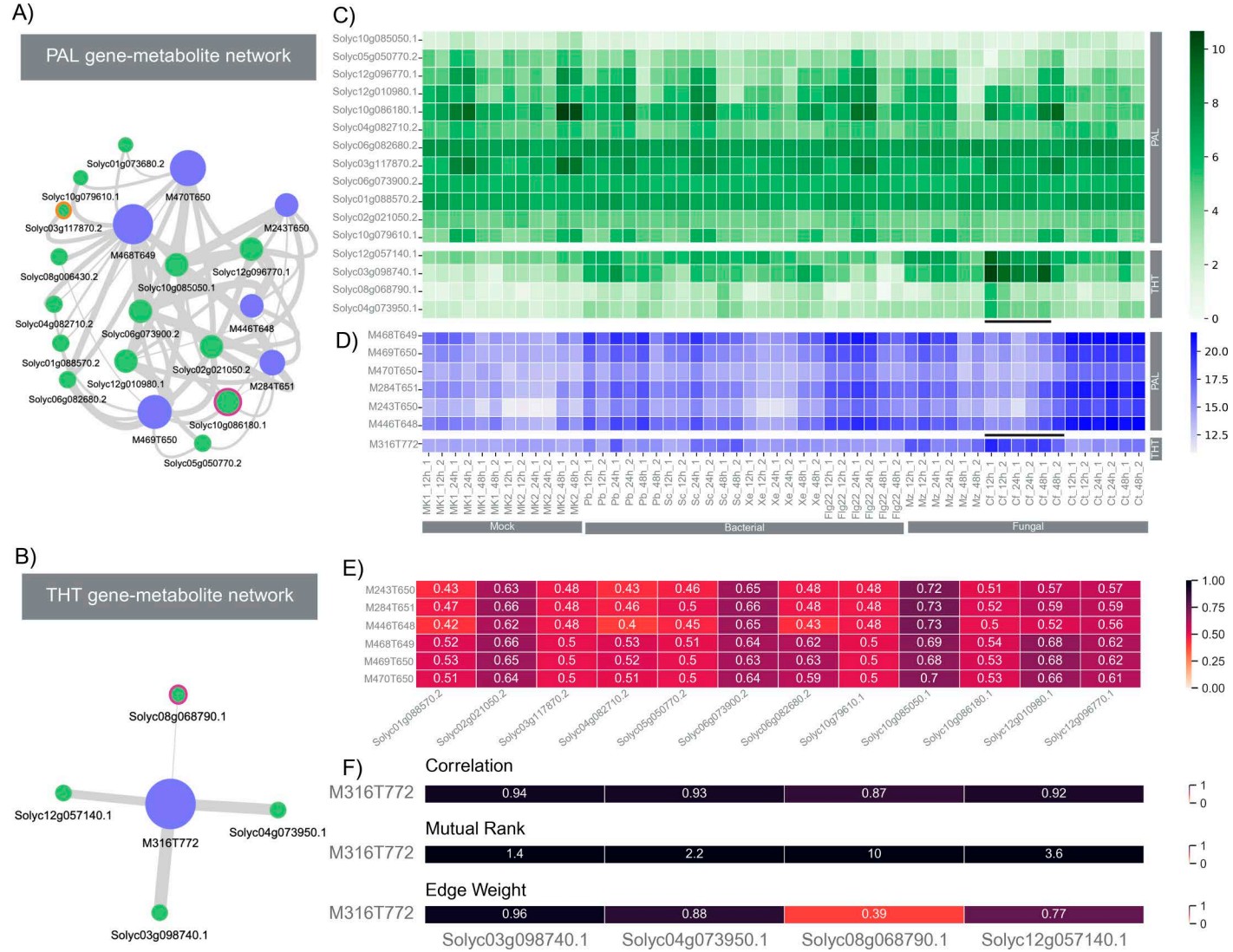

**Fig 4. Detection of functional clusters (FCs) specific to the phenylalanine (PAL) and p-coumaroyltyramine (THT) pathways. (A)** Network depicting the relationship between transcripts and mass signatures within the PAL FC. The data underlying the PAL network graph can be found at https://zenodo.org/records/15697913. Edge scores values were extracted from the table *Jeon_bac_correlations_MR_weights_DR_25* (Jeon_bacterial.sqlite) for the features present in the table *Jeon_bac_correlations_clone_DR_25 with the* cluster ID number 77. **(B)** Network illustrating the interplay between transcripts and mass signatures within the THT FC. The data underlying the PAL network graph can be found at https://zenodo.org/records/15697913. Edge scores values were extracted from the table *Jeon_fungal_correlations_MR_weights_DR_10* (Jeon_fungal.sqlite) for the features present in the table *Jeon_fungal_correlations_clone_DR_10 with the* cluster ID number 120. **(C)** Heatmap illustrating the expression levels of all transcripts within the PAL and THT FCs. The raw data underlying the transcripts heatmap can found at https://zenodo.org/records/15697913/files/rnaseq.rpkm.csv. **(D)** Heatmap displaying the abundance of all mass signatures present in the PAL and THT FCs. The raw data underlying the metabolite heatmap can be found at https://github.com/sattely-lab/falcarindiol_pathway_metabolomics. **(E)** Correlation matrix highlighting the correlations among transcripts and mass signatures within the PAL FC. **(F)** Correlation matrix displaying the relationships between transcripts and mass signatures within the THT FC, including Mutual rank and transformed edge weights.The data underlying the correlation matrices of PAL and THT and can be found at https://zenodo.org/records/15697913.

In another biosynthetic pathway, namely the α-tomatine pathway, we observed the presence of genes distributed across multiple FCs (S5 Fig). This pathway involves nine specific biosynthetic genes responsible for converting cholesterol into α-tomatine, and these genes have been extensively characterized in tomato [20]. MEANtools captured all biosynthetic genes involved in the glycoalkaloid metabolism (GAME) group, including GAME1, GAME4, GAME6, GAME7, GAME11, GAME12, GAME17, and GAME18, in nine different FCs (S9 Fig). Furthermore, GAME9, an APETALA2/Ethylene response factor, related to regulator of the steroidal glycoalkaloid pathway in tomato, was also captured within one of the 9 FCs. Additionally, we found biosynthetic genes involved in the synthesis of precursors for the α-tomatine pathway, such as SQS (Squalene Synthase), TTS1 (β-Amyrin Synthase), TTS2 (β-Amyrin Synthase), and SSR2 (Sterol Side Chain Reductase 2), present in multiple instances throughout the network. We used coexpression network to merge FCs, resulting in coexpression edges between biosynthetic genes from the α-tomatine pathway GAME12 transaminase, and 2-oxoglutarate-dependent dioxygenase GAME11, and GAME17 (UDP-glucosyltransferase) and GAME1 (UDP-galactosyltransferase) (S8 Fig). Additionally, MEANtools successfully pinpointed another crucial biosynthetic gene associated with the chlorogenic acid biosynthetic pathway, known as HQT (Hydroxycinnamoyl-CoA quinate: hydroxycinnamoyl transferase). HQT plays a pivotal role in facilitating the transformation of quinic acid into caffeoyl quinic acid, which represents another specialized metabolite within the phenylpropanoid pathway.

### MEANtools facilitates prioritization of reaction steps using reaction likelihood scores

MEANtools generates reaction-likelihood scores based on substrate-enzyme association, for each anticipated reaction. To obtain the score, the likelihood of each atom in the substrate is calculated for being a site-of-metabolism using the GNN-SOM [44] method. This results in an array of likelihoods for each atom in the substrate. Later, using ReactionDecoder [45], reaction centers and bond cleavages are predicted between each substrate and product (Fig 5A and 5B). MEANtools makes use of this information to extract likelihood scores only for atoms that are involved in reaction centers and bond cleavages. The maximum value of likelihood score within the reaction center represents the reaction likelihood score. Fig 5C shows the distribution of likelihood scores for experimentally characterized enzyme-substrate pairs, referred to as *Known* in S2 Data, and randomly assembled enzyme-substrate pairs as *Random*. The likelihood scores differ significantly (Mann-Whitney U statistic: 2573.0, P-value: 4.4e-07) between Known and Random pairs, with median and mean values of 0.86 and 0.70 for Known pairs, and 0.29 and 0.39 for Random pairs, respectively.

## Discussion

MEANtools can generate testable hypotheses on metabolic pathways with little to no prior knowledge, by integrating metabolomics and transcriptomics data. This method effectively automates the identification of key Pfam domains required for a specific reaction and allows users to tune the reaction-Pfam domain associations according to their level of confidence or based on the taxa of origin. To do so, MEANtools queries RetroRules, a retrosynthesis-oriented enzymatic reactions database, showing that tools and methods within the retrosynthetic biology and synthetic pathway design fields have considerable application potential for metabolic pathway prediction and potentially SM discovery. To enable meaningful transcript-metabolite associations, high-quality transcript annotations are of high importance, particularly to ensure correct identification of Pfam domains relevant to biosynthesis. We emphasize that transcriptome data not only supports data integration but also plays a vital role in improving gene annotations, especially for biosynthetically active genes. This can be achieved using tools like BRAKER [46], along with co-expression analysis that helps to pinpoint candidate genes involved in metabolic pathways. Furthermore, if comprehensive protein annotations and abundance data across samples are available, proteomic datasets can serve as an alternative to transcriptomic data, at least for the correlation part. Slight modifications are required to map the proteomics data with the RetroRules database.

Metabolomics and transcriptomics datasets are typically used as CSV-formatted pre-processed tables featuring mass-feature abundances and transcript expressions, respectively. Integrating such datasets solely through

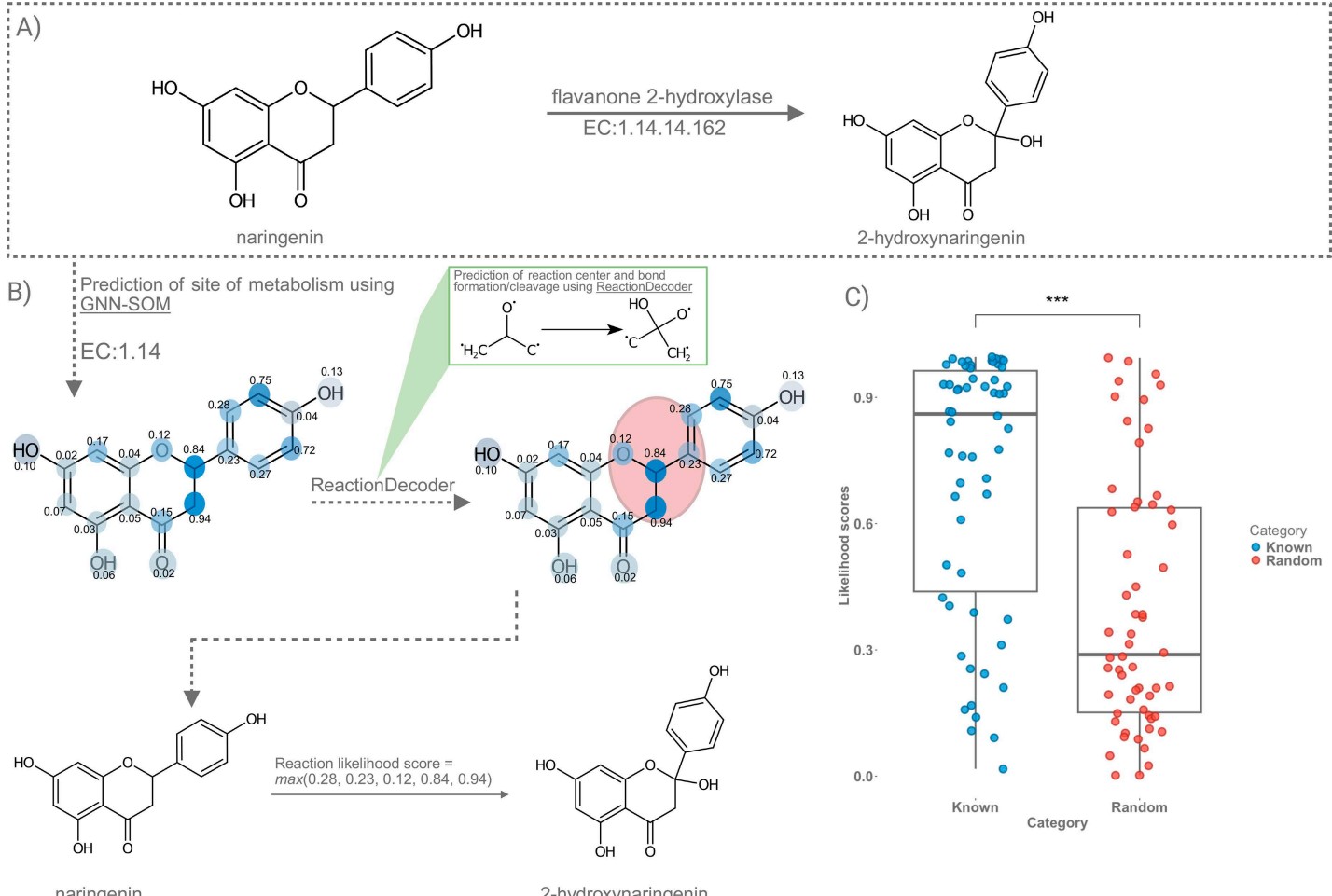

**Fig 5. Overview of the estimation of reaction likelihood scores. (A)** The transformation of naringenin to 2-hydroxynaringenin requires a flava-
none 2-hydroxylase enzyme. **(B)** To estimate the likelihood score of this reaction, the SMILES ID and the enzyme EC number was used as an input
to the GNN-SOM method. GNN-SOM predicts likelihood scores of each atom in the molecule for being a site-of-metabolism. As a next step, we take
the SMILES ID of the substrate and the product and use the ReactionDecoder tool to identify the reaction centers and possible bond formation/cleav-
age site(s). Referring to the atom index of the atoms in the reaction center, we select the highest likelihood score. This value represents the reaction
likelihood score of a reaction which is 0.94 for the transformation of naringenin to the 2-hydroxynaringenin. **(C)** Distribution of reaction likelihood scores
from experimentally validated enzyme-substrate pairs (Known) and randomly assigned enzyme–substrates pairs (Random). The raw data underlying the
distribution of likelihood scores can be found in S2 data.

Pearson-based correlations often results in many false-positive associations. Additionally, determining an optimal thresh-
old for eliminating weak correlations poses significant challenges. The use of mutual-rank statistics has proven effective
for constructing global gene co-expression networks, as demonstrated by Wisecaver *and colleagues* [14]. Leveraging this
approach, we utilized the MR-based method to develop a correlation-based global gene-metabolite network. This network
highlights strongly correlated genes and metabolites. Ideally, individual FCs should advance to the next stage of pathway
prediction. However, the FCs size may sometimes be insufficient for forming a complete biosynthetic pathway. The FCs
size proved stable across the treatment combinations in Jeon *and colleagues* (2020) dataset [30] (S7 Fig). Given that
genes and metabolites in plant biosynthetic pathways tend to overlap, FCs are also overlapping in nature. MEANtools
provides a script (*merge_clusters.py*) to merge multiple FCs that share common mass features. Mass features that exhibit

distinct abundance patterns across samples are then grouped into separate clusters following this merging process. This step is crucial for ensuring enough mass features and transcripts remain to either fully or partially reconstruct a biosynthetic pathway. Changing the size of FCs is also possible using the ClusterONE [47] inbuilt parameter. However, this also changes the clustering pattern of mass features and transcripts. Additionally, the current method to provide significance to each FC could also be improved, as this was originally developed for co-expression datasets.

The RetroRules database is publicly available as SQLite database and can be used directly with MEANtools. The three reaction rules datasets resulting from RetroRules, *loose*, *medium*, and *strict* are available in a single CSV file in the https://zenodo.org/records/15697913. MEANtools includes these three different datasets as an input parameter (*strict, medium,* and *loose*, respectively) to allow the user to constrain the predictions for specific purposes and find the right balance between sensitivity and specificity, considering the tradeoff between enzymatic annotation confidence and diversity of the resulting set of enzymatic reactions. In the *strict* dataset, which is a smaller subset of reaction-rule-enzyme associations, the number of resulting candidate genes in the final reaction anticipations was reduced due to its more specific Pfam annotations. On the contrary, reaction anticipations with the *loose* set were associated to unrelated Pfams (S7 Fig), such as AminoTran_1_2 or Glyco_transf_20, responsible for transferring amino and sugar groups, respectively [48]. Such unrelated Pfams were not found in the *strict* rule dataset, as shown in the hydroxylation of octadecadiene-diynoic acid into metabolite 6 and its subsequent hydroxylation into metabolite 7. These spurious links, coming from RetroRules, have been kept in the *loose* dataset after applying the Pfam cutoff of 6, which highlights the importance of using the strict rules dataset when specificity is preferred over sensitivity. Most importantly, both predictions correctly predict the enzyme associated to the conversion of octadecadiene- diynoic acid into metabolite 6, as reported by Jeon *and colleagues* (2020) [30]. Transient expression of this enzyme in *Nicotiana benthamiana* (Solyc10g100250) was experimentally associated to depletion of crepenynic acid and the production of two new metabolites [30]. According to the observed LC–MS profile, one of the metabolites was putatively identified as octadecadiene-diynoic acid, making plausible the role of Solyc10g100250 in its conversion to metabolite 6. By cross-checking these reaction-enzyme association datasets with sets of correlated enzyme-coding genes and metabolites, MEANtools effectively filters the set of possible mass shift-reaction associations based on the available -omics evidence.

MEANtools refines biosynthetic pathway predictions by integrating transcriptomic and metabolomic data through a robust correlation-based framework. By employing mutual-rank statistics [14,49], it enhances the interpretation of correlation values through incorporation of expression rank consistency across samples. As illustrated in Fig 4, this approach allows for the subtle interpretation of transcript-metabolite relationships within the PAL and THT FCs. For example, although multiple THT transcripts show strong correlation with the metabolite M316T772, their expression dynamics vary not only between the transcripts but also across time points. The MR score effectively captures these variations, helping to distinguish the most biologically relevant associations. The transcript Solyc08g068790.1 exhibits a high Pearson correlation (0.87) but compared to other genes its edge weight with M316T772 is relatively low (0.39), reinforcing its time-based (12 h post-infection) and sample-specific (*C. fulvum*) expression dynamics in the same biosynthetic process. This refined scoring mechanism enhances MEANtools ability to prioritize candidate transcripts and metabolites supporting more precise hypotheses generation in pathway discovery. In the case of PAL and THT biosynthetic pathways, the reconstruction of reaction steps using the prediction step of MEANtools was hindered due to two main factors. Firstly, the conversion of phenylalanine to p-coumaroyl-CoA involves stereoisomers, which are not captured by the mass spectrometric data. Secondly, the conversion to p-coumaroyltyramine requires two substrates, tyrosine, and p-coumaroyl-CoA, whereas RetroRules-based rules are designed for single-substrate reactions only. Although RetroRules contains a rule for the stereomeric conversion of phenylalanine to p-coumaroyl-CoA, MEANtools filters such rules involving stereomeric structures to avoid complexity.

The initial construction of the substructure map occurs once, either during the initial retrieval of the RetroRules database or during updates (step 1, Fig 1B). MEANtools uses this substructure map to generate pathway predictions. To

this end, in step 2 (Fig 1C), it predicts possible metabolites and their corresponding molecular structures for each mass feature by identifying possible adducts and querying the LOTUS database, or a user-defined metabolite database that can be supplied in CSV format. According to the IUPAC recommendation 2013 [50], adduct ions are formed when a molecule combines with another ion or neutral species during the ionization process in mass spectrometry. This results in a shift in the observed m/z (mass-to-charge) value, which differs from the molecule's neutral mass. Common adducts include protonation ($[M + H]^+$), sodium attachment ($[M + Na]^+$), or loss of water ($[M - H_2O]^+$) among others. These variations are instrument-dependent and can significantly influence metabolite identification. To address this issue, users can input adducts known to be common in the mass spectrometry platform. Alternatively, MEANtools uses a curated default list of ~40 commonly observed adducts, both for positive and negative mode, compiled from the study by Huang and colleagues [51]. The refined m/z values, after adjusting for most formed adducts, are then mapped against LOTUS database. MEANtools structural mapping can benefit with resolution MS1 data, such as that generated by Orbitrap or time-of-flight mass spectrometers. These instruments provide the mass accuracy necessary for confident adduct-based metabolite matching and molecular formula estimation. Given that the resolution of the mass spectrometer directly impacts the precision of detected *m/z* values. MEANtools includes a tunable parameter to define the mass accuracy threshold in parts per million (ppm). By default, the system operates at a 20 *ppm* tolerance, which offers a balance between sensitivity and specificity. However, users working with ultra-high-resolution instruments may adjust this parameter (e.g., to 5 ppm) to reduce less significant structural assignments. This flexibility allows MEANtools to accommodate a broad range of instrument capabilities and experimental designs.

The LOTUS database was converted to an SQLite format to be compatible with MEANtools, and it was made available in the GitHub repository. In step 3 (Fig 1E–G), MEANtools exclusively queries reactions that would yield metabolites with mass features that can be mapped within the metabolome or as ghost mass signatures. Collectively, this strategic approach enables MEANtools to efficiently utilize computing resources when generating *in-silico* molecules. Because of step 3, MEANtools produces a sequence of subsequent reactions, along with predicted products for all pairs of mass signatures, correlated enzyme-coding genes, and references to characterized reactions and enzymes that served as the rules for predicting these reactions. Step 3 can be iterated multiple times, as desired by the user, enabling the generation of pathway predictions extending beyond a single enzymatic reaction away from the initial query molecule.

Because of MEANtools' flexible and modular design, there is room for improvement in many of its processing steps. Annotating mass signatures with predicted structures can be improved by allowing to load MS/MS data and use mass spectral library and networking-based annotation approaches [52] to increase accuracy and allow validation, in a similar way as done by MetWork [38]. Gene-metabolite clusters can further be improved by a more elaborate co-expression and/or molecular network analysis. Converting predicted reaction networks into DAGs is currently used to study and present unsupervised predictions, but more complex manipulations of the network may allow for predictions better tailored for the user, such as prioritizing specific reactions or molecular substructures, for example, by integrating MS2LDA analyses [53]. We also note that further curating the reaction-Pfam domain associations or allowing the user better control over them by allowing customized reaction-rule databases could improve the method as well: some enzyme domains may be linked to large numbers of reactions, likely leading to false positives when the objective is to predict biosynthesis pathways, but these enzyme domains could be useful when exploring the biosynthetic potential of a structure when designing a synthetic pathway. Finally, the reaction likelihood scores can also be improved by adopting or developing precise methods for reaction site predictions.

Altogether, we present a novel computational method to predict metabolic pathways guided by multi-omics evidence, allowing researchers to conveniently generate testable and easy-to-browse hypotheses. Furthermore, we anticipate that our work provides the basis for future work to expand the numbers of ways in which paired genomic, transcriptomic, and metabolomic data can be used to link natural product chemistry to biosynthesis genes and producers, and to analyze biosynthetic diversity in nature.

PLOS Biology

## Methods

### Correlation-based integration generates testable associations

Global reconstruction of co-expression modules in gene expression data has been shown to be a powerful method to identify groups of genes involved in the same metabolic pathway when querying for modules with genes that encode biosynthetic enzymes [14]. In MEANtools, instead of generating co-expression modules using transcriptomics dataset, FCs are generated for different network sizes by integrating transcriptomics and metabolomics data (Fig 1D). Inspired from the work of Wisecaver *and colleagues* [14], correlation values between mass features and transcripts are first converted to MRs [49], which are then subjected to an exponential decay function that converts continuous MR values to numbers between 0 and 1 and referred here as *edge weights*. Both node types and edge weights are further subjected to clustering using ClusterONE [47], which results in multiple overlapping FCs. Each FC represents a significant association of mass abundance and transcript expression patterns across samples. In a network view, mass feature and transcripts represent two unique node types connected by edge weights. MEANtools allows users to visualize the expression of each FC in the form of heatmaps with transcripts sorted in three categories according to the protein domains they encode, following the same categorization used by plantiSMASH: scaffold-generating enzymes, tailoring enzymes, and the remaining genes [10].

### Rescaling input data using Median Absolute Deviation (MAD)

MEANtools employs the Median Absolute Deviation (MAD) for data rescaling. MAD calculates the median of all values within the dataset, which represents the 50th percentile. It then determines the absolute difference between each value and the calculated median and ensures that the differences are expressed as positive values, regardless of whether they are greater or less than the median. Finally, computing the median of these absolute differences yields the MAD (equation 1).

$$MAD = median( \, | \, X_i - median(X) \, | \, )$$

(1)

where $X_i$ refers to the $i_{th}$ row element present in the data matrices.

### Computation of mutual rank and edge weights

Pairwise correlations of transcripts and mass features are converted to MRs (calculated as a geometric mean of the rank of Pearson correlation coefficient (PCC) of transcript A to mass feature A and of the PCC rank of mass feature A to transcript A. This MR statistic is calculated for every transcript-mass feature pair. Since the MR value can vary between 1 and $n - 1$, where $n$ represents the total number of features in either the transcriptomic or metabolomic dataset, we transform the MR scores into *edge weights*, ranging between 0 and 1, using an exponential decay function. By default, MEANtools computes edge weights by using four different rates of decay (5, 10, 25, 50) resulting in five different networks of varying sizes (equation 2). The modified exponential decay function is:

$$N_{i \rightarrow j} : e^{-(MR-1)i \rightarrow j}$$

(2)

where i→j refers to multiple DR and $N_{i \rightarrow j}$ represents a combined network generated using $i \rightarrow j$ DR. MR denotes the estimated MR between genes and metabolites. Gene–metabolite pairs that show lower edge score than 0.01 are excluded in the $N_{i \rightarrow j}$ networks.

MEANtools then employs the graph-clustering method ClusterONE [47], which identifies overlapping clusters of transcripts and mass features. Clustered transcripts and mass features can assemble into a biologically significant subnetwork, which we refer to as a FC. Such clusters represent a higher-level organization of the transcriptome and metabolome. The average number of FCs per network decreases with increasing network size. For each FC, ClusterONE assigns

a *p*-value derived from the comparison between edges within the FC and those that radiate out of the FC. The resulting network, stemming from various DR, is then stored within an SQLite database.

## Mass-shifts associated to reactions serve as templates for pathway predictions

MEANtools leverages the established relationships between reactions and their associated mass shifts to scan the input metabolome. It assigns molecular structures to each mass feature of the metabolome by mapping them with a list of adducts and then querying LOTUS database (downloaded on 10/10/2023). LOTUS database was converted to an SQLite format to be compatible with MEANtools and made available in the GitHub repository. It identifies pairs of mass features with discernible differences in mass-charge ratios that can be logically explained by known reactions. Within this process, one mass feature is annotated as a potential substrate, while the other is marked as a product. It is worth noting that a given mass shift might be assigned to more than one reaction, and many reactions are bidirectional in nature. Consequently, any pair of mass features can be associated with multiple reactions, considering both forward and reverse directions. Additionally, recognizing that not all metabolites within a metabolic pathway may reach detectable levels in the (measured) metabolome, MEANtools optionally generates 'ghost mass signatures.' These ghost signatures serve as virtual, unmeasured intermediates between any two metabolites that possess measured mass signatures. This concept, recently introduced in MetWork [38] in the construction of metabolic networks based on MS/MS spectra, is also applied here. Notably, although the ghost mass feature is provided as an option to use for all reactions, it automatically gets switched on when MEANtools fails to assign mass features either as substrates or products. By incorporating information on reaction-mass-shift associations, MEANtools constructs a comprehensive reaction network. This network comprises mass signatures connected by annotated reactions and forms the foundational framework for the subsequent prediction of metabolic pathways.

## Prediction of metabolic pathways

MEANtools leverages the reaction network to facilitate the generation of pathway predictions. Initially, it predicts potential metabolites along with their corresponding molecular structures for each mass signature. Subsequently, MEANtools employs the RDKit [54,55] Python package (v 2019.03.2.0) to computationally generate *in silico* structures resulting from each reaction-associated substrate.

Given the substantial number of reactions cataloged in RetroRules, generating all product molecules for the metabolite structures predicted in a metabolome by querying every reaction can be time-consuming and computationally intensive. From each reaction, new metabolites emerge, leading to a large number of molecular structures. To expedite this process, MEANtools relies on -omics evidence, specifically the reaction-substrate-enzyme pairs under the confinement of FCs, to guide the generation of *in silico* molecules.

Further acceleration is achieved by targeting specific substructures within each metabolite structure, employing a divide-and-conquer strategy (Fig 6). For each metabolite structure, MEANtools initiates by verifying the presence of specific atoms, such as N or C. Upon success, the next step involves querying reactions that pertain only to simple substructures, like N=N and C=C. If both atoms are present, MEANtools extends its search to reactions centered on substructures like C=N and C–N. In subsequent rounds, MEANtools explores more complex substructures based on the substructures identified in prior steps. For instance, metabolites with the C=N substructure are exclusively queried for reactions centered on the C=N-C substructure. This iterative process continues until no further successful queries are obtained for a given metabolite.

## Easy-to-browse MEANtools output

MEANtools generates user-friendly visualizations and supplementary data in the form of easy-to-browse tables. MEANtools stores these tables in an SQLite database. It also offers python-based utility scripts to retrieve and visualize FCs within the MR-based correlation network.

**Fig 6. MEANtools identifies reactions for a molecular structure according to a divide-and-conquer strategy.** For each metabolite, MEANtools first queries the presence of key atoms and then continues to query, in rounds, increasingly complex reactant substructures according to which substructures have already been identified. For example, a set of metabolites is first queried for nitrogen and carbon atoms. Metabolites that pass these criteria are then queried for more complex substructures like C–N or C=C. In the following round, MEANtools queries substructures with more complexity according to which substructures have already been identified: in this manner, only metabolites with the N=C substructure is queried for the N=C–N substructure.

MEANtools analyzes the reaction network created in the preceding stages to predict candidate metabolic pathways aligned with the user's interests. To accomplish this, the NetworkX [55] Python package (v2.4) is utilized. MEANtools constructs a distinct subnetwork for each of the initial metabolites provided by the user. These subnetworks are transformed into DAGs by identifying any cycles within the network, representing potential reversible reactions. Only links capable of advancing the reaction away from the initial metabolite are retained. In instances where cycles occur among metabolites at the same reaction distance from the initial metabolite, the edge featuring the weakest enzyme-metabolite correlation is eliminated. This approach yields multiple DAGs rooted at the initial metabolites, each offering the potential for candidate metabolic pathways. The longest reaction path in each subnetwork, commencing from the initial metabolite, is identified to predict these pathways. This process is repeated to generate a DAG for each initial metabolite at the termination of the reaction, yielding two pathway predictions for each input structure. MEANtools then delivers the complete reaction network and all DAGs in the form of CSV tables, facilitating seamless import and exploration within Cytoscape. Furthermore, pathway predictions are presented as SVG image files, providing comprehensive details regarding the involved metabolites, reactions, genes, and their respective correlations. To enhance user exploration, MEANtools offers an option to generate SVG files for each molecular structure predicted in earlier stages. This lets users pinpoint and prioritize structures or reactions of interest. MEANtools can construct DAGs and pathway predictions rooted at any user-selected molecule.

## Supporting information

**S1 Fig. Venn diagram of the content of three datasets, namely loose, medium and strict.** *Loose* dataset contains reaction rules-enzyme associations from the RetroRules database, cross-referenced with the Rhea and KEGG-orthology database. *Medium* dataset contains experimentally validated entries together with the ECDomain-Miner predictions. The *strict* dataset contains only experimentally validated entries. The raw data underlying the Venn diagram can be found at https://zenodo.org/records/15697913/files/Combined_small_middle_big_datasets..csv
(DOCX)

**S2 Fig. Distribution of taxonomic groups in the reaction-enzyme loose dataset.** X- and Y-axis represent categories of the taxonomic group and their counts, respectively. The raw data underlying the taxonomic distribution can be found at https://zenodo.org/records/15697913/files/Combined_small_middle_big_datasets..csv
(DOCX)

**S3 Fig. Distribution of *p*-values from the correlation analysis between the processed transcriptomics and metabolomics datasets from Jeon and colleagues, 2020 [32].** The data underlying the *p*-values distribution can be found at https://zenodo.org/records/15697913.
(DOCX)

**S4 Fig. Distribution of correlation coefficients from the correlation analysis between the processed transcriptomics and metabolomics datasets from Jeon and colleagues, 2020 [30].** The data underlying the distribution of correlation coefficients can be found at https://zenodo.org/records/15697913.
(DOCX)

**S5 Fig. Functional clusters (FC) encompassing genes from the alpha-tomatine pathway of *Solanum lycopersicum*.**
(DOCX)

**S6 Fig. Prediction of intermediate steps of falcarindiol pathway from MEANtools using *strict* (A) and *loose* (B) datasets.** Number of enzyme associations is reduced while using *strict* dataset due to its more specific Pfam annotations. Blue arrow shows non-specific enzyme associations predicted with *loose* dataset.
(DOCX)

**S7 Fig. The figure illustrates a comparison of the sizes of various functional clusters (FC) across a range of decay rates and treatments.** The X-axis represents individual decay rates, categorized into treatments involving bacteria, fungi, or a combination of both. The Y-axis denotes the size of the FC, which includes both gene and metabolite features. The results indicate that larger cluster sizes are associated with higher decay rates and larger sample sizes. The data underlying the distribution of FCs size versus decay rates can be found in S5 Data.
(DOCX)

**S8 Fig. Use of coexpression edges to merge and prioritize FCs.** Green edges represent coexpression networks and blue edges represent gene-metabolite networks of FCs. Colored nodes represent biosynthetic genes annotated from seven tomato pathways. Coexpression was detected across all treatment dimensions. Coexpression networks and FCs were created with a mutual rank metric and ClusterONE clustering with a decay rate of 10. FCs from the α-tomatine pathway are connected thanks to coexpression edges between the genes GAME12 transaminase (Solyc12g006470), and 2-oxoglutarate-dependent dioxygenase GAME11 (Solyc07g043420), and GAME17 (UDP-glucosyltransferase) (Solyc07g043480) and GAME1 (UDP-galactosyltransferase) (Solyc07g043490). Two genes associated with the biosynthesis of 4-coumarate CoA ligase (4CL) were also connected by coexpression edges in the hydroxy cinnamic acid amide (HCAA) pathway. Merging FCs via coexpression edges between biosynthetic genes was robust across decay rates 10

and 25, with only the connections belonging to the HCAA pathways displayed in decay rate 5. The data underlying this network graph can be found in S6 Data.
(DOCX)

**S9 Fig. Biosynthetic pathway of *p-coumaroyltyramine* [44].**
(DOCX)

**S1 Data. Classification of plant biosynthetic genes. Table presents a classification of biosynthetic genes based on profile Hidden Markov Models (pHMMs), including PFAM accessions, enzyme names, EC numbers, and their source organisms.**
(XLSX)

**S2 Data. Collection of plant-based biosynthetic reactions. This dataset lists selected biosynthetic reactions, from Singh and colleagues, Fig 1 [8] with associated substrates, products, enzyme EC numbers, and SMILES structure.** It also includes reaction likelihood scores for known, random, and falcarindiol-related reactions.
(XLSX)

**S3 Data.** *m/z*-Based annotation of mass-features. Table provides *m/z*-based annotations of falcarindiol-related mass-features.
(XLSX)

**S4 Data. Predicted reactions.** Table contains predicted biosynthetic reactions from MEANtools, detailing root mass-features, corresponding predicted products, mass transitions, and associated correlated transcripts.
(XLSX)

**S5 Data. Combined list of functional clusters.** Table lists the functional clusters, from four networks 5, 10, 25, and 50, identified through correlation analysis using bacterial, fungal, and combined datasets [30]. Differentially expressed transcripts were selected by comparing each treatment group to the mock, as well as the combined treatment groups to the mock. These genes were then used to perform correlation analysis, resulting in the generation of functional clusters.
(XLS)

**S6 Data. List of coexpression edges.** Table consists of co-expression edge weights calculated between all genes in the transcriptomic dataset [30]. The edge weights were generated using transcriptomic data with the parameter *-ft* of the *corrMultiomics.py* script
(XLS)

## Author contributions

**Conceptualization:** Kumar Saurabh Singh, Hernando Suarez Duran, Marnix H. Medema.

**Data curation:** Kumar Saurabh Singh, Hernando Suarez Duran, Olga Zafra Delgado.

**Formal analysis:** Kumar Saurabh Singh, Hernando Suarez Duran, Elena Del Pup, Olga Zafra Delgado.

**Funding acquisition:** Saskia C. M. van Wees, Marnix H. Medema.

**Investigation:** Kumar Saurabh Singh, Hernando Suarez Duran, Elena Del Pup.

**Methodology:** Kumar Saurabh Singh, Hernando Suarez Duran, Olga Zafra Delgado.

**Project administration:** Saskia C. M. van Wees, Marnix H. Medema.

**Software:** Kumar Saurabh Singh, Hernando Suarez Duran, Olga Zafra Delgado.

**Supervision:** Saskia C. M. van Wees, Justin J. J. van der Hooft, Marnix H. Medema.

**Validation:** Kumar Saurabh Singh, Hernando Suarez Duran.

**Writing – original draft:** Kumar Saurabh Singh, Hernando Suarez Duran.

**Writing – review & editing:** Elena Del Pup, Olga Zafra Delgado, Saskia C. M. van Wees, Justin J. J. van der Hooft, Marnix H. Medema.

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
