## [Editor Report · Decision Letter 0]

Dear Dr Medema,

Thank you for submitting your manuscript entitled "MEANtools: multi-omics integration towards metabolite anticipation and biosynthetic pathway prediction" for consideration as a Methods and Resources Article by PLOS Biology.

Your manuscript has now been evaluated by the PLOS Biology editorial staff, as well as by an academic editor with relevant expertise, and I am writing to let you know that we would like to send your submission out for external peer review.

Once your full submission is complete, your paper will undergo a series of checks in preparation for peer review. After your manuscript has passed the checks it will be sent out for review. To provide the metadata for your submission, please Login to Editorial Manager (https://www.editorialmanager.com/pbiology) within two working days, i.e. by Feb 14 2025 11:59PM.

Kind regards,

Richard

Richard Hodge, PhD

rhodge@plos.org

PLOS

---

## [Decision Letter · Decision Letter 1]

Dear Dr Medema,

Thank you for your patience while your manuscript "MEANtools: multi-omics integration towards metabolite anticipation and biosynthetic pathway prediction" went through peer-review at PLOS Biology. Please accept my sincere apologies for the delays that you have experienced during the peer review process. Your manuscript has now been evaluated by the PLOS Biology editors, an Academic Editor with relevant expertise, and by two independent reviewers.

In light of the reviews, which you will find at the end of this email, we are pleased to offer you the opportunity to address the comments from the reviewers in a revision that we anticipate should not take you very long. We will then assess your revised manuscript and your response to the reviewers' comments with our Academic Editor aiming to avoid further rounds of peer-review, although we might need to consult with the reviewers, depending on the nature of the revisions.

As you will see, the reviewers are positive about the combined transcriptomic and metabolomic approach for metabolic pathway prediction. Reviewer #1 raises concerns about the use of transcriptomes and asks for clarification for why proteomic data was not used, as well as clarifying if data from time-of-flight mass spectrometry is required for estimating the structures of the metabolic compounds. In addition, Reviewer #2 notes that the additional explanations should be included for how the approach can refine existing biosynthetic hypotheses, as well as integrating the code into the online tools to enhance utility.

We expect to receive your revised manuscript within 2 months. Please email us (plosbiology@plos.org) if you have any questions or concerns, or would like to request an extension.

**IMPORTANT - SUBMITTING YOUR REVISION**

*Resubmission Checklist*

*Published Peer Review*

*PLOS Data Policy*

*Blot and Gel Data Policy*

Best regards,

Richard

Richard Hodge, PhD

rhodge@plos.org

REVIEWS:

Reviewer #1: The manuscript describes a computational system referred to as MEANtools to predict metabolic pathways based on the reaction rules and metabolite structures stored in public databases. The approach of using transcriptomics and metabolomics data for metabolic pathway prediction would be novel. The validity of this approach is illustrated in a case study involving falcarindiol biosynthesis. However, the use of transcriptomes means that prediction accuracy will decrease if gene annotations are not correct. Currently, annotation curation is still required, so it is questionable whether this method will be widely used.

Other comments:

1. The amount of transcripts does not necessarily correlate with the amount of translation. The authors need to explain why this study uses the transcriptome rather than the proteome.

2. The authors mentioned that MEANtools can assign potential structure matches by identifying adducts in the metabolome and querying the LOTUS database for matching metabolites based on molecular weight. It would be better if there was a detailed explanation of what kind of compound data is being used. For example, what exactly is an adduct? There are various types of adducts of ions detected, and multiply charged ions should also be detected.

3. When estimating the structures of metabolic compounds, it may also be necessary to consider the analytical accuracy required of the mass spectrometer. Will data from a time-of-flight mass spectrometer with high mass accuracy be required?

4. On line 268, the authors mention that MEANtools also anticipated steps five and six of the pathways. Has a system capable of predicting metabolic pathways that include many steps like this been reported? If not, the authors may wish to emphasize this point as an advantage of their research.

5. PAL, 4CL, THT are abbreviations. Therefore the full name should be used the first time it appears in the text body. In addition, it should be better to show the metabolic pathway including PAL, 4CL and THT.

Reviewer #2: The authors developed MEANtools, a computational tool designed for metabolomics and systems biology studies. This tool integrates transcriptomics (gene expression data) and metabolomics (mass spectrometry data) to identify and predict metabolic pathways, providing a powerful framework for exploring biosynthetic processes.

In systems biology research, the integration of omics data is crucial for understanding complex metabolic interactions, yet this manuscript presents an important contribution by offering clear guidance on how transcriptomic and metabolomic datasets can be effectively combined.

While the manuscript is well-written, it could benefit from further clarity in explaining how the the relationship between transcripts and mass signatures modifies or refines existing biosynthetic hypotheses, particularly as illustrated in Figure 4.

The use of mass shift to predict chemical reactions has also been used by Jourdan et al in 2008 (doi:10.1093/bioinformatics/btm536), could be interesting to add this reference.

The presented code works well but its integration into one of the multiple online tools developed by the authors could help future users.

Based on these comments, I would recommend the acceptance of this work.

---

## [Editor Report · Decision Letter 2]

Dear Dr Medema,

Thank you for your patience while we considered your revised manuscript "MEANtools: multi-omics integration towards metabolite anticipation and biosynthetic pathway prediction" for publication as a Methods and Resources Article at PLOS Biology. This revised version of your manuscript has been evaluated by the PLOS Biology editors and the Academic Editor.

Based on our Academic Editor's assessment of your revision, I am pleased to say that we are likely to accept this manuscript for publication, provided you satisfactorily address the following data and other policy-related requests that I have provided below (A-E):

(A) We routinely suggest changes to titles to ensure maximum accessibility for a broad, non-specialist readership. In this case, we would suggest a minor edit to the title, as follows. Please ensure you change both the manuscript file and the online submission system, as they need to match for final acceptance:

“MEANtools is a computational pipeline that integrates multi-omics data to identify metabolites and predict biosynthetic pathways”

(B) You may be aware of the PLOS Data Policy, which requires that all data be made available without restriction: http://journals.plos.org/plosbiology/s/data-availability. For more information, please also see this editorial: http://dx.doi.org/10.1371/journal.pbio.1001797

-Supplementary files (e.g., excel). Please ensure that all data files are uploaded as 'Supporting Information' and are invariably referred to (in the manuscript, figure legends, and the Description field when uploading your files) using the following format verbatim: S1 Data, S2 Data, etc. Multiple panels of a single or even several figures can be included as multiple sheets in one excel file that is saved using exactly the following convention: S1_Data.xlsx (using an underscore).

-Deposition in a publicly available repository. Please also provide the accession code or a reviewer link so that we may view your data before publication.

Figure 2B-D, 4C-E, S2, S3, S4, S7

(C) Please also ensure that each of the relevant figure legends in your manuscript include information on *WHERE THE UNDERLYING DATA CAN BE FOUND*, and ensure your supplemental data file/s has a legend.

(D) Please note that we cannot accept sole deposition of code in GitHub, as this could be changed after publication. However, you can archive this version of your publicly available GitHub code to Zenodo. Once you do this, it will generate a DOI number, which you will need to provide in the Data Accessibility Statement (you are welcome to also provide the GitHub access information). See the process for doing this here: https://docs.github.com/en/repositories/archiving-a-github-repository/referencing-and-citing-content

(E) Please ensure that your Data Statement in the submission system accurately describes where your data can be found and is in final format, as it will be published as written there.

We expect to receive your revised manuscript within two weeks.

*Published Peer Review History*

*Press*

Best regards,

Richard

Richard Hodge, PhD

rhodge@plos.org

PLOS

---

## [Editor Report · Decision Letter 3]

Dear Dr Medema,

On behalf of my colleagues and the Academic Editor, Matthew Wook Chang, I am pleased to say that we can accept your manuscript for publication, provided you address any remaining formatting and reporting issues. These will be detailed in an email you should receive within 2-3 business days from our colleagues in the journal operations team; no action is required from you until then. Please note that we will not be able to formally accept your manuscript and schedule it for publication until you have completed any requested changes.

PRESS

Best wishes, 

Richard

Richard Hodge, PhD

rhodge@plos.org

PLOS
